# Absolute Quantification in Diagnostic SPECT/CT: The Phantom Premise

**DOI:** 10.3390/diagnostics11122333

**Published:** 2021-12-11

**Authors:** Stijn De Schepper, Gopinath Gnanasegaran, John C. Dickson, Tim Van den Wyngaert

**Affiliations:** 1Department of Nuclear Medicine, Antwerp University Hospital, 2650 Edegem, Belgium; tim.vandenwyngaert@uza.be; 2Faculty of Medicine and Health Sciences (MICA—IPPON), University of Antwerp, 2610 Wilrijk, Belgium; 3Department of Nuclear Medicine, Royal Free London NHS, London NW3 2QG, UK; gopinath.gnanasegaran@nhs.net; 4Institute of Nuclear Medicine, University College of London Hospitals NHS, London NW1 2BU, UK; john.dickson2@nhs.net

**Keywords:** absolute quantification, SPECT/CT, phantoms, diagnostics, 3D printing

## Abstract

The application of absolute quantification in SPECT/CT has seen increased interest in the context of radionuclide therapies where patient-specific dosimetry is a requirement within the European Union (EU) legislation. However, the translation of this technique to diagnostic nuclear medicine outside this setting is rather slow. Clinical research has, in some examples, already shown an association between imaging metrics and clinical diagnosis, but the applications, in general, lack proper validation because of the absence of a ground truth measurement. Meanwhile, additive manufacturing or 3D printing has seen rapid improvements, increasing its uptake in medical imaging. Three-dimensional printed phantoms have already made a significant impact on quantitative imaging, a trend that is likely to increase in the future. In this review, we summarize the data of recent literature to underpin our premise that the validation of diagnostic applications in nuclear medicine using application-specific phantoms is within reach given the current state-of-the-art in additive manufacturing or 3D printing.

## 1. Introduction

From the moment radioactivity was discovered by Henri Becquerel in 1896, there was an immediate interest in quantification arising from the need to study this phenomenon. After more than a century, this interest in measuring and quantifying radioactivity is more relevant than ever within the context of SPECT/CT technology in nuclear medicine. The past two decades have seen the continuous improvement of SPECT/CT as a molecular imaging modality to a point where it can produce accurate quantitative images [1,2,3]. The technical advances leading to this improved accuracy have been extensively reviewed before [4,5].

Recent research on absolute quantification has primarily focused on applications related to theranostics or radionuclide therapy [6,7,8,9,10] and dosimetry [11,12,13,14,15,16,17]. Next to its use in the therapeutic setting, accurate quantification of tracer uptake could become highly relevant, providing diagnostic information beyond just the absence or presence of disease. Until now, relative quantification and comparisons against a database have dominated quantitative applications of SPECT/CT in diagnostic studies. However, while the quantitative capability of SPECT/CT as a modality is without doubt and several potential applications have already been suggested at the beginning of the previous decade [18], there is still limited use in clinical practice.

The results from a survey on the use of quantitative SPECT in the UK in 2019 show that quantitative SPECT/CT has not yet broken through in clinical practice [19]. Approximately two-thirds (67%) of responders indicated not using absolute quantification or a semiquantitative standardized uptake value (SUV). The main focus of those who do use absolute quantification is radionuclide therapy, predominantly for thyroid conditions and neuroendocrine tumors. While most respondents indicated using quantitative images only for therapy, 43% indicated having a calibration for ${}^{99m}$Tc. When asked what the main impediment is for quantitative SPECT, 35% questioned the benefit, and 23% indicated a lack of transferability across sites and platforms. Clearly, several challenges, including clinical validation and transferability, continue to hamper the use of absolute quantification for diagnostic purposes.

Three-dimensional printing or additive manufacturing is a technique where a structure is built layer by layer, and each new layer is deposited on the previous layer [20]. Also called rapid prototyping, it was initially developed for making scale models of a prototype, which at that time were still developed by skilled craftsmen based on 2D drawings. The printers use a 3D drawing which it subdivides into the individual layers to be constructed. There is a large flexibility in methods and materials that are used for printing. Fused deposition modeling (FDM) and stereolithography (SLA) are among the most popular commercial 3D printing technologies. A wide variety of materials can be used to print, from metals like aluminum to polymers such as poly(methyl methacrylate) (PMMA). In recent years, improvements in 3D printing technology have decreased the cost while increasing the speed with which the prints are produced. This allows for efficient prototyping and iterative designs where improvements to the model are introduced with each iteration. The high quality in printing technology is reflected in the homogeneity within prints and the reproducibility of the prints. The availability of the technology combined with its reproducibility has the added benefit that phantoms can be reproduced at different sites rather than having to send a phantom from one site to the other for multicentre studies. These favorable properties have increased the interest in 3D printing for a wide variety of applications, including anthropomorphic phantoms for medical imaging [21]. In this narrative review, we summarize the available applications reported in the literature, focusing on the potential of innovative models or phantoms to improve the clinical uptake of this technology. We will show that these applications of 3D printing have already shown they can fulfill the need for application-specific anthropomorphic phantoms. These phantoms have already had a significant impact on our understanding of the physical effects that can potentially lower the accuracy of quantitative SPECT/CT imaging. It is our premise that they will further increase our understanding and allow for optimization of application-specific protocols.

## 2. Understanding the Need for Quantification

The concept of theranostics in nuclear medicine is well-established and increasingly successful, especially with the introduction of ${}^{177}$Lu-based therapies [7,8,10]. Undeniably, the measurement of the absolute activity in this context is a requirement for accurate dose calculation, as these start from the number of nuclei decaying in a region of interest. Afterward, the decay energy and range of the emitted particles are used to calculate the absorbed dose. Establishing a dose-effect relationship for a given radionuclide therapy allows for a personalized therapy plan more closely resembling external beam radiotherapy. Moreover, such dosimetry calculation after radionuclide therapy has also become a legal requirement in the European Union (EU) under the Basic Safety Standard (Council Directive 2013/59/Euratom of 5 December 2013).

The use of absolute quantification in diagnostic applications is perhaps less of a requirement than in therapeutic uses. Today, uptake ratios and reference databases are routinely used, with applications in cardiac amyloidosis [22], temporomandibular joint growth [23], dopaminergic function [24], and renal function [25]. More (semi)quantitative metrics have also been popularized, of which the standardized uptake value (SUV) has been the most popular. There are several variants of SUV, but in essence, they represent a normalization of the activity in a region-of-interest by dividing the injected activity by some patient metric of distribution volume, such as body mass. There are several diagnostic applications where an increased tracer uptake is correlated with the presence or severity of the disease. Multiple authors have investigated the link between the uptake of ${}^{99m}$Tc-labeled bone-seeking agents and whether a bone lesion is benign or malignant [26,27,28,29]. In cardiac imaging, the uptake of a ${}^{99m}$Tc-labeled bone-seeking agent has been shown to correlate with current clinical standard quantitative measures for amyloid burden in cardiac amyloidosis, therapy response, extracellular volume as measured on NMR, and left ventricular mass index, as measured on echocardiography [30]. Together, these preliminary applications illustrate that further progress in absolute quantification in diagnostic applications can significantly increase the value of these imaging studies as a truly quantitative biomarker of disease severity, patient outcomes, or predictor of treatment response.

## 3. Requirements for Absolute Quantification

Accuracy and reproducibility are essential for applying absolute quantification in diagnostic medicine, which depend on technical and physical processes but also on the patient’s biology and physiology. Mirroring the considerable efforts over the last decades to improve our understanding of the biological and technological factors influencing the SUV in PET imaging [31,32,33] and the need for standardization [34], SPECT/CT has seen similar progress, even though a number of challenges remain.

The major physical processes challenging the accuracy and reproducibility for diagnostic applications are photon attenuation and scatter and the limited spatial resolution of SPECT/CT cameras. Bone structures or metallic implants will strongly affect the image produced due to their increased density. The limited spatial resolution is caused by the collimator and the detector not being perfect systems. Many structures in the human body are very small or have an irregular shape, and depending on the application, corrections for this phenomenon need to be considered. However, while many techniques have been reported, few methods, if any, are available for routine clinical use [35].

Apart from the physical processes, the technical parameters related to image reconstruction should be optimized to guarantee sufficient accuracy and reproducibility for a given application. Different vendors use different proprietary reconstruction and correction algorithms, including, but not limited to attenuation correction [36], scatter correction [37], collimator modelling [38], and resolution modelling [35]. The parameters of these algorithms need to be validated and standardized to result in accurate and reproducible (semi)quantitative metrics.

Finally, the biokinetics of the tracer being studied can impact quantification as well. For example, variations in the clearance of activity from the background can occur (e.g., depending on renal function), and tracers may demonstrate variable washout from the target over time, influencing the estimation of uptake in a target lesion. In theory, each patient is ideally scanned multiple times to establish a time–activity curve from which the tracer kinetics can be derived, providing information on the biokinetics depending on patient biology and physiology. While some suggestions have been made for multiple time-point imaging in diagnostics [39,40], generally, only one image at a fixed time-point is available in the clinical routine. In practice, the choice of imaging time-points is usually informed by the population biokinetics of the tracer of interest [24,41,42,43,44] to minimize the biological variability and increase reproducibility [45,46].

## 4. From Feasibility Study to Clinical Practice

There are several lessons to be learned from the experience with PET/CT when translating feasibility studies to clinical practice. The biological variance in biokinetics and technical variance have to be well-controlled to standardize the acquisition and analysis. In SPECT, contrary to PET, only a minimum incubation period after injection is usually considered, and there is no standardization in the incubation window after injection. In bone scintigraphy, a minimum of 2 h of incubation is recommended according to the EANM guideline [42], while in the FDG PET guideline from EANM, a recommendation of imaging 55 min to 75 min post-injection is made [41]. There is currently a large variety of technology available, ranging from analog planar NaI-crystal cameras to digital circular geometry solid-state cadmium-zinc-telluride (CZT) designs. These technologies differ in sensitivity, energy resolution, spatial resolution [47,48]. Therefore, results from quantitative analysis can differ significantly, and it is difficult to translate from one technology to another.

Considerable efforts have been made to progress towards more standardization in SPECT/CT imaging. Several initiatives have attempted to extend the approach from the EARL program for FDG PET/CT [49] to SPECT/CT [11,13,47], its main advantage being the large availability of the specific phantoms within the nuclear medicine community: a cylinder with a volume of 5–7 L and the NEMA IEC phantom. First, a cross-calibration is performed using the cylinder. The cylinder is filled with a known volume of water and known activity measured in a radionuclide calibrator. The activity concentration as recovered in the SPECT/CT is compared to the known activity concentration by dividing the known volume of the cylinder and the known activity from the radionuclide calibrator measurement. The use of a large cylinder allows for a measurement without boundary interactions and related partial volume effects. Second, the NEMA IEC phantom consists of a large volume, considered as background, with spherical inserts of different sizes, considered the volumes-of-interest (VOIs). The spheres are filled with a different activity concentration than the background region. The activity concentration as recovered in the SPECT/CT is again compared to the known activity concentration. Through the partial volume effect, the activity in the spheres will deviate more for the smaller than the larger spheres. The recovery coefficients will also deviate as a function of contrast between spheres and background, as spill-in and spill-out will present differently. In general, lower contrast will result in lower recovery of the activity in the smaller spheres [50].

The translation of diagnostic SPECT/CT applications in cardiac amyloidosis or bone growth across different technologies is also hampered by a lack of validation. While it makes sense in oncological applications, such as FDG PET/CT, to use the activity recovered in a hot or cold sphere in a background volume as validation, this is not necessarily the case for applications with very different geometry. Considering the uptake of ${}^{99m}$Tc-labeled bone-seeking agents in the myocardium, the challenges for accurate quantification are very different. The myocardium differs from a spherical shape, and there is a large volume of background activity adjacent to it. The partial volume effect (PVE) will differ when measured on cameras with different spatial resolutions or different reconstruction algorithms. Depending on the situation, there might be spill-in from the hot background into the colder VOI or spill-out from the hot VOI into the colder background affecting the results and the clinical translation.

Validation of the absolute activity concentration requires verification of the measured activity concentration. Except for the bladder, where the activity concentration in the urine can be measured afterward, such in vivo verification is most often impossible in humans [51]. Therefore, a reasonable alternative is the development of application-specific phantoms.

## 5. Phantoms in Nuclear Medicine

From the onset of nuclear imaging [52] and emission computed tomography [53], phantoms have been used to test image properties and imaging techniques. The quest for possible clinical applications required validation of the technology, and phantoms could provide such solid foundation. Phantoms are designed to reflect the situation under study as closely as possible, such as a cylinder with a sleeve of activity and two different-sized spherical inserts to investigate lesion contrast in brain imaging [53].

Ideally, phantoms should mimic our patients as closely as possible, a quality referred to as anthropomorphism. The most relevant properties for nuclear medicine are photon–matter interaction, and geometry and activity distribution. Fortunately, humans mostly consist of water, which is reflected in the photon attenuation coefficient of human tissue. Water and plastics, such as PMMA, which have similar attenuation coefficients, are popular materials for soft tissue applications. However, geometry is essential as patients or study populations can have very different anatomies. An obvious example is children versus adults, but also between adults in, for example, patients with very different BMI, or the density of healthy bone versus osteoporotic bone, which influences attenuation. Different pathologies and tracers can also have a very different tracer distribution in the same organ of interest. For example, a tracer for myocardial perfusion will have different kinetics and distribution than a bone-seeking agent repurposed for cardiac amyloidosis.

Several commercial phantoms are available for a human torso [54,55], spine [56], and brain [57]. These phantoms are realistic and can be used to measure the recovered activity concentration in the relevant regions. They are, however, limited to whole organ volumes, but pathology does not necessarily affect the entire organ. For example, cardiac amyloidosis deposition can be heterogeneous in the cardiac wall [58]. In that case, amyloid deposits should show as hot spots within the cardiac wall. It is immediately apparent that validation of absolute quantification for this application requires a more pathology-specific approach, and the limited number of commercially available anthropomorphic phantoms may not necessarily be appropriate for all applications. An anthropomorphic phantom has the potential to take the specific anatomical situation into account, including the depth at which the region of interest lies within the patient, but can also take almost any type of material into account. We can imagine bone structures near the region of interest or self-attenuation when bony structures are the region of interest. While water has roughly the density of soft tissue, it can be made more dense and equivalent to different types of bone by adding K${}_{2}$HPO${}_{4}$ to accommodate these requirements [59]. For tissue types with lower density than water, this is more difficult. The activity in the lungs is contained in water, while they are mostly filled with air. This duality presents a challenge for anthropomorphic lung phantom production, as it is difficult to reduce the density of water to that of air. A fillable 3D-printed anthropomorphic phantom will provide with the previously mentioned exception an anatomically realistic representation, and allow for flexibility in scanning conditions and experimental designs by varying filling fluids to measure the impact of nearby material on attenuation and scatter.

Designing a phantom has long been cumbersome and expensive. Their complex shapes require a sufficient level of detail and the flexibility to adapt the phantom for a different patient or cohort while not sacrificing production speed or increasing cost. Additive manufacturing or 3D printing is a technique that can achieve this. A summary of several properties making 3D printing an attractive technigue for phantom production, as well as an example of such a printed shape are shown in Figure 1 This technique has been widely applied to produce patient-specific implants and is starting to make its way into medical imaging.

## 6. 3D-Printed Phantoms: A New Hope?

Recent technological progress has drastically lowered the cost and infrastructure needs for 3D printing, spreading its use in applied research, including nuclear medicine. In the case of phantom production, this allows more simplified development of purpose-specific phantoms. This technology makes it possible to produce almost any shape conceivable with relative ease. Several studies using 3D-printed phantoms in nuclear medicine have recently been published Table 1 for applications in the following anatomical regions: the abdomen [60,61,62], pancreas and kidneys [63], kidneys [64,65], brain [66,67,68], head and neck [69], systolic and diastolic heart [70], lungs [71], and patient-derived lesion shapes [11]. With current technology, anthropomorphic phantoms can be 3D-printed with a precision in the µm-range, well beyond the sub-millimeter resolution of CT or MRI used to develop these anatomical models.

As a proof-of-concept, an abdomen phantom was first developed by Gear et al. [62], including the liver, kidneys, and spherical inserts. The assessment of the phantoms included measuring concordance of the geometry with that of imaging-derived organ dimensions, which showed a maximum deviation of 7%, and a visual assessment of PET and SPECT acquisitions.

Afterwards, they created an abdomen phantom [60] for validation of quantitative imaging of ${}^{99m}$Tc and ${}^{90}$Y and dosimetry after liver radioembolization. The phantom included a fillable liver and lesions in a solid abdomen. The design process was discussed in detail, including considerations on attenuation, leak-tightness, and attachments. The included flowchart should provide sufficient information to reproduce a similar phantom. The 3D-printed object must match the anatomical structure as closely as possible, especially for smaller objects, to evaluate the partial volume effect accurately. For the liver, the deviation of the volume was 9.6%, even though the contours on the original MRI images were hand-drawn and were thoroughly smoothed before printing to remove pixelation. This agreement was deemed sufficient for the desired application. The X-ray properties were characterized at different photon energies for several different materials and were similar to soft tissue. The accuracy of the recovered activity was different for the lesions and liver, and very different for various isotopes on PET or SPECT. Deviations as high as 18% were observed for the liver and more than 60% for the lesions. The difference is probably caused by the partial volume effect. The authors also included the cost of their project (approximately €11,600 adjusted for inflation), demonstrating the increased affordability of 3D printers today (costing €1000–3000). At the time of writing, the phantom had undergone approximately 20 acquisition protocols, including at different sites, which illustrates the excellent durability.

Robinson et al. [61] developed an abdomen phantom to verify activity quantification in SPECT in the context of molecular radiotherapy using ${}^{177}$Lu, including ${}^{99m}$Tc imaging. The phantom included simplified models used in a dosimetry system common at that time (OLINDA/EXM), representing spleens and kidneys of different sizes, and also representing a pancreas and liver of the ages of 5 years, 10 years, and that of an adult (extended with a bilobar design and tumor insert). The description of the phantom properties is brief, yet information on attachments and filling ports can be useful for other projects. The prints were of similar accuracy compared to the virtual volumes as in the previous examples. They evaluated the accuracy of the quantification by estimating the calibration factor, which is the ratio of the activity recovered in the VOI over the injected activity. In the ideal measurement, this would be 1. As expected, the calibration factor decreased with decreasing organ volume. Importantly, a clear dependency of the calibration factor on organ shape could be demonstrated for ${}^{99m}$Tc and ${}^{177}$Lu, resulting in a reduction of the absorbed dose for the liver, spleen, and kidneys using organ-specific factors. This finding illustrates that non-spherical calibration factors from 3D-printed phantom inserts can significantly improve the accuracy of whole organ activity quantification.

The challenges of imaging pancreatic beta cells using ${}^{111}$In-exendin, which includes high uptake in the nearby kidneys, motivated Woliner van der Weg et al. [63] to develop pancreatic and kidney phantom inserts for the NEMA IEC phantom. They compared several reconstruction algorithms and different activity concentrations in the pancreas. Visually, the images were comparable to real-life studies performed in humans and showed comparable artefacts. With this work, they could assess the need for different correction algorithms and determine the appropriate reconstruction settings most suitable for clinical use.

Tran-Gia et al. first developed a single-compartment kidney model for dosimetry [64] similar to the mathematical model from MIRD Pamphlet 19 [72] for newborns, one-year olds, five-year olds, and adults, and later, a two-compartment kidney model [65] by dividing the previous volumes into a cortex (70%) and a medulla (30%). The design and production of the single-compartment model were described in detail, including the software used and attachments. A volume assessment showed a maximum deviation of 5.8%. They compared different reconstruction algorithms and different PVC techniques to optimize imaging for ${}^{177}$Lu radionuclide therapy. In this setup, the difference between spherical and renal recovery coefficients suggests that a more geometry-specific alternative should replace the typically applied volume-dependent lookup tables based on spherical recovery coefficients. The cost of the project was similar to that by Gear et al. [60] in the order of magnitude of €10,000.

Iida et al. [66] developed a brain phantom to simulate a static cerebral blood flow distribution in the grey matter, while including a realistic skull structure. They described the segmentation upon which the phantom was based in detail. The grey matter and skull structures were fillable with liquid. White matter was made from the polymer used by the printer. The phantom was printed in five-fold and evaluated for geometry and attenuation using CT. The volumes of all phantoms were consistently close to the original value with a maximum deviation of 2.6%. The phantom was filled with either ${}^{99m}$Tc or ${}^{123}$I for SPECT or ${}^{18}$F for PET. The images were compared with a virtual image blurred using a Gaussian filter to simulate the lower resolution of SPECT and PET. The printing of hollow structures normally requires supporting pillars, yet the authors were able to forgo this requirement by reducing the temperature, speed, and pitch of the printer.

A different approach to a brain phantom was developed by Negus et al. [67] by dividing the head and brain from an MRI image into slabs. These slabs with a thickness of 4 mm are 3D-printed for attenuation purposes. A conventional inkjet printer with a cartridge with added ${}^{99m}$Tc was used to print a greyscale image of grey matter on a sheet of paper. The appropriate paper was placed between the slabs. The phantom is shown in Figure 2. The attenuation was measured using transmission scans, and the images were visually assessed.

The previously mentioned brain phantoms focus on grey and white matter, while imaging of the dopamine receptors also plays a significant role in neurological imaging. Jonasson et al. [68] produced a brain phantom with striatal inserts of different sizes to evaluate the influence of the size on the interpretation of PET imaging. The information on the design and production of the phantom is scarce. The background volume of the phantom and the striatum was filled with differing concentrations of ${}^{18}$F. The recovery coefficients were compared for different reconstruction algorithms, yielding a much better recovery for all sizes of striatum using OSEM combined with the point-spread function (PSF). The PSF technique should increase the spatial resolution of the image and therefore decrease the PVE. The large increase in recovery coefficient by adding PSF to the OSEM reconstruction demonstrates this.

The head and neck region is very complex where tiny structures are densely packed, yet this region is often imaged in the context of sentinel lymph node (SLN) detection or thyroid disease. A phantom was designed and tested by Alqahtani et al. [69]; however, the analysis was aimed at assessing the imaging ability of gamma cameras for distinguishing different SLNs and the thyroid, and no evaluation of the absolute quantification was performed.

The left-ventricular ejection fraction (LVEF) is a function of the end-diastolic and end-systolic heart volume. Verrechia-Ramos et al. [70] printed the end-diastolic and end-systolic phase of the same heart based on a gated cardiac MRI, but provided little information on the design and the production process of the phantom. As the reliability of the volumes is the only important factor in determining the LVEF, no evaluation of the absolute activity measurement was done. Different isotopes and imaging modalities are used to evaluate the LVEF: MRI, planar scintigraphy (${}^{99m}$Tc), SPECT (${}^{99m}$Tc), and PET/CT (${}^{18}$F). An interesting outcome was the feasibility of a very short PET/CT acquisition for a first-pass FDG scan to evaluate the LVEF.

For the validation of the accuracy of the quantification procedures in the SEL-I-METRY trial, Gregory et al. [11] developed three 3D-printed lesion inserts based on a sub-carinal node but of different sizes. The validation was performed using 123I and 131I. Little information is provided on the phantom production, nor the resulting accuracy of the quantification in the lesions.

Motion artefacts are common in molecular imaging of the thorax due to respiratory motion, and decrease the detectability and hamper quantification. Black et al. [71] developed a functional anthropomorphic lung phantom using 3D-printed molds. At the time of writing, there is no information available on the resulting images and quantification.

Recent developments attempt to improve some shortcomings of phantoms. For example, the two-compartment kidney model was complex to assemble and required the preparation of two stock solutions. Theisen et al. [73] recently presented work on a single-compartment kidney phantom. This phantom has two regions with different spatial densities by the introduction of gyroid structures. These gyroids are surfaces that limit the volume that can be filled using the stock solution. By having a denser gyroid for the medulla than for the cortex, the activity concentration will be different, even though the entire phantom is filled with the same stock solution.

There are also successful attempts at 3D-printing radioactive geometric phantoms [74,75,76]. The main advantage of the latter is the absence of cold walls and, when implanted with longer-lived isotopes such as ${}^{68}$Ge (half-life = 271 days), could be used for validation in multi-center PET trials. These phantoms could play a role in quality control. However, there remains discussion on their utility as for some, this would increase the capability for diverse quality control. In contrast, according to others, they would increase the complexity while moving the difficulty in phantom preparation from the stock solution to the stock solution and the quality control of the 3D printer [77].

## 7. Discussion

Even though 3D printing has distinctive advantages, there is always a learning curve associated with any new technique. This learning curve could be further reduced by including more information on the design process of the phantoms and the choices made in publications. For example, Gear et al. [62] and Tran-Gia et al. [64] described the design process in more detail, including information on the attachments. This is essential information for the production of the phantom which can be used by others when designing their own phantoms. Several authors included information on wall thickness, which is useful to guarantee leak tightness. All authors included information on what printer, material, and software were used. This information all contributes to a faster uptake of the methodology in other centers. It is, however, a worrying trend that more recent publications compared to earlier work provide less information necessary for the reproduction, or to push the innovation of phantoms forward. A recommendation regarding key information to be included in future publications on 3D-printed phantoms is summarized in Table 2. It would be of considerable value if publishers and/or scientific societies would establish an online repository similar to the NIH 3D Print Exchange [78] and actively encourage making models available upon publication of research in their respective journals. The emphasis placed on data availability in the context of validation, reanalysis, and reproduction of medical research should extend to availability of phantoms.

From the articles that mention project cost, a trend in better affordability is evident. While the price five years ago was still approximately €10,000, 3D printers are now available from as low as €1000–3000.

For anthropomorphic phantoms, it is essential that they accurately represent the anatomical image on which they are based. Therefore, verification of the geometry and the attenuation are essential. Several authors have compared the 3D-printed phantoms to the original volumes. Only small deviations from the original volumes were observed, which proves the reliability of the technique, and supports their use when reproduced given the importance of geometry on quantification. The same applies to the attenuation coefficients of the materials used. It is established that most polymers used in 3D printing have attenuation coefficients similar to soft tissue and can be used as such. Water can be made to have similar attenuation coefficients as bone by adding K${}_{2}$HPO${}_{4}$. It is difficult to reduce the density and attenuation coefficients of water to those of air. Therefore, anthropomorphic phantoms of the lungs are notable challenges as the activity in the lungs is present in water, while the organ mostly consists of air.

While phantoms for kidney dosimetry started as simple geometric phantoms, the complexity is still increasing with every new generation. This single-compartment phantom was improved to a two-compartment model (Figure 3). Today, even more advanced anthropomorphic phantoms and innovative designs for varying tracer concentrations within a single compartment phantom are possible. Future challenges include the 3D printing of molds for elastic anthropomorphic phantoms that can be used for dynamic imaging.

The applications of some of the discussed phantoms have already changed our view on quantitative SPECT/CT. The kidney phantom by Tran-Gia and Lassmann has been used to evaluate quantification [65], kidney dosimetry [14] using ${}^{177}$Lu, eventually extending it to a multicentre setting [79]. The main finding of their work was that the typical volume-based approach for partial volume correction based on spherical inserts like in the IEC NEMA Body Phantom was insufficient for accurate quantification. Geometry plays an important role in the accuracy, and this should be reflected in the evaluation of specific applications. The evaluation of voxel-based dosimetry has taught us to use caution when applying this technique due to the large difference with the true measurement. This problem was improved by application of a specific partial volume correction software, indicating that quantitative imaging has to be optimised for every application. The multicentre evaluation showed a large variety in initial performance, but also the potential for harmonisation for this application.

Until now, the major applications of 3D-printed phantoms of absolute quantification have been in the context of dosimetry. Apart from the striatal phantom, all other diagnostic applications have been evaluated visually. Yet, from these, we have learned to understand the importance of small and irregular geometries. For example, the hot spots in ${}^{111}$In-Exendin imaging of the pancreas can result from increasing the iterations of the reconstructions and are not necessarily a result of heterogeneous uptake of the tracer. Even though there was no evaluation of the quantitative accuracy of the reconstruction, it allowed for optimization of the acquisition for visual interpretation.

## 8. Conclusions

Several examples have been discussed for the possible application of absolute quantification in SPECT/CT. There has, however, not been a broad uptake of these applications in diagnostic nuclear medicine so far. The establishment of absolute quantification in clinical practice depends on validation of the accuracy and reliability, and application-specific validation can benefit from anthropomorphic phantoms tailored to the application. The continued advances and availability of 3D printing allow for such application-specific phantoms to be developed at a reasonable cost and in a reasonable time with relative ease. With the current state-of-the-art, we have seen increasing possibilities and increasing complexity in the designs. These innovative designs allow for more realistic phantoms with every subsequent generation. 3D-printed phantoms have already changed our perspective on the limitations of absolute quantification while providing the possibility for further improvements. They will increase the opportunities to validate the application of absolute quantification in SPECT/CT and increase the acceptance of absolute quantification in clinical practice. This is The Phantom Premise.

## Figures and Tables

**Figure 1 diagnostics-11-02333-f001:**
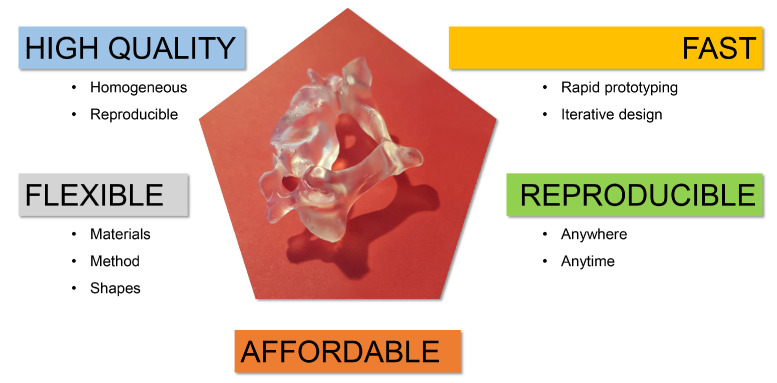
Summary of some properties that make 3D printing suitable for anthropomorphic phantom production. In the centre is a 3D-printed model of a cervical vertebra (C3). You can appreciate the level of detail in the processus spinosus, the processus transversus and the foramen transversarium, for example.

**Figure 2 diagnostics-11-02333-f002:**
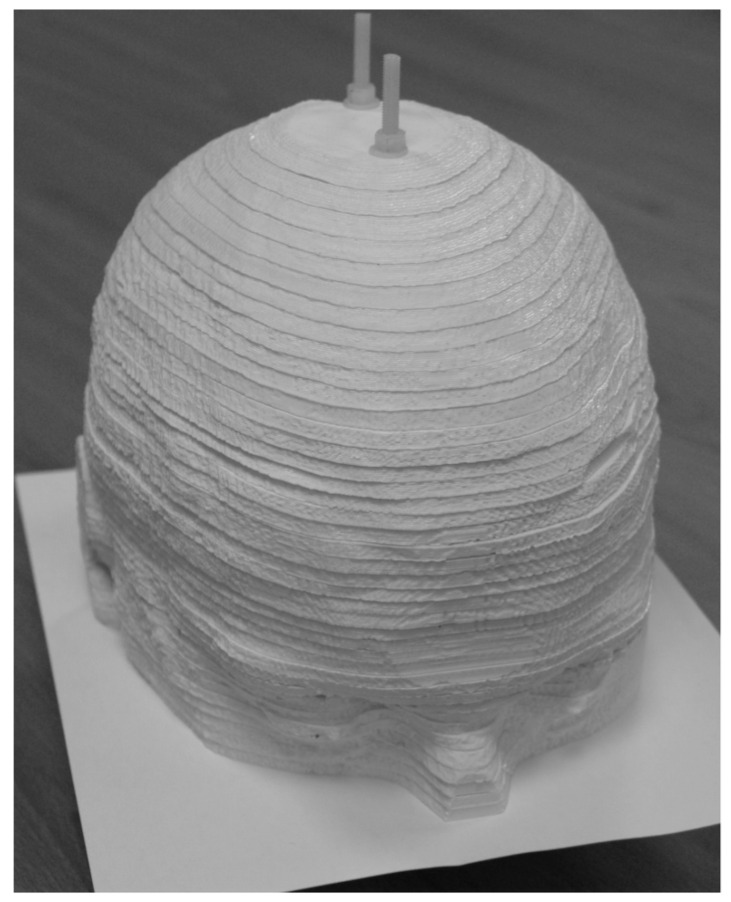
The brain phantom developed by Negus et al. [67] follows a different approach compared to fillable phantoms. It is a sandwich of 3D-printed slabs for attenuation and paper for activity distributions. The image was reproduced under license from the American Association of Physicists in Medicine and John Wiley & Sons, Inc. (Hoboken, NJ, USA).

**Figure 3 diagnostics-11-02333-f003:**
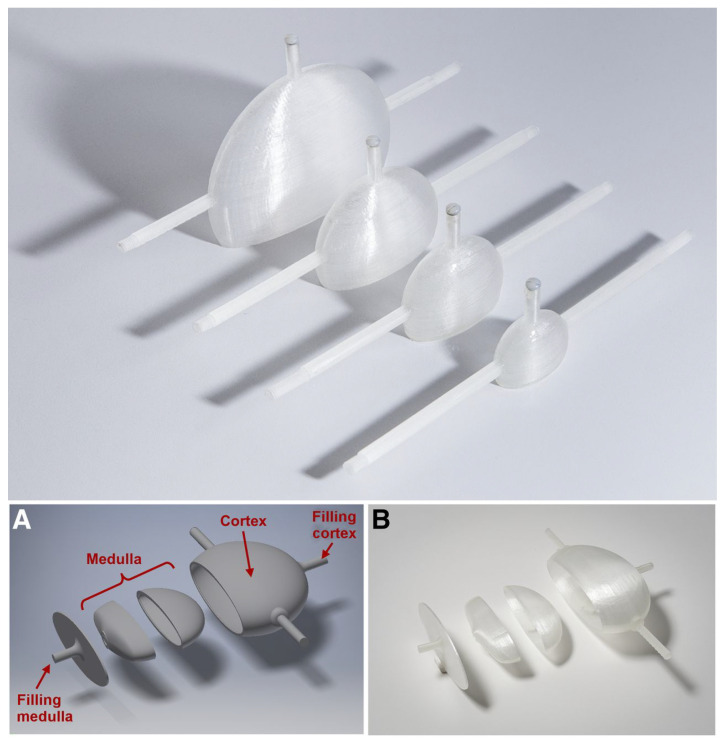
Three-dimensional printing allows for an iterative design process where improvements can be made for a next-generation print. The example shows the single-compartment kidney produced by Tran-Gia et al. [64] (**top**) and the two-compartment kidney phantom (**A**,**B**) subsequently produced by the same research group, Tran-Gia et al. [65] (**bottom**). This research was originally published in JNM by Tran-Gia et al. [64,65]. *©* SNMMI.

**Table 1 diagnostics-11-02333-t001:** Summary of all articles and phantoms included in this review. SLA = stereolitography, FDM = Fused Deposition Modelling, CNR = contrast-to-noise ratio, RC = recovery coefficient, CF = calibration factors, LVEF = left ventricular ejection fraction.

Author	Region	Method	Evaluation	Isotopes
Iida et al., 2013 [66]	Brain	SLA	Visual	${}^{18}$F, ${}^{99m}$Tc, ${}^{123}$I
Gear et al., 2014 [62]	Abdomen (liver, spleen, kidneys)	SLA	Visual	${}^{18}$F, ${}^{99m}$Tc
Gear et al., 2016 [60]	Liver, spherical inserts	FDM	Total activity	${}^{99m}$Tc, ${}^{90}$Y
Negus et al., 2016 [67]	Brain	FDM	Visual	${}^{99m}$Tc
Tran-Gia et al., 2016 [64]	Kidney	FDM	CF	${}^{99m}$Tc, ${}^{177}$Lu, ${}^{131}$I
Tran-Gia et al., 2018 [65]	Kidney	FDM	CF	${}^{177}$Lu
Robinson et al., 2016 [61]	Abdomen (liver, spleen, kidney, pancreas)	FDM	CF	${}^{99m}$Tc, ${}^{177}$Lu
Woliner-van der Weg et al., 2016 [63]	Pancreas, kidney	FDM	Ratio	${}^{111}$In
Alqahtani et al., 2017 [69]	Head & Neck	FDM	CNR	${}^{99m}$Tc
Jonasson et al., 2017 [68]	Striata	FDM	RC	${}^{18}$F
Verrecchia-Ramos et al., 2021 [70]	Heart	FDM	LVEF	${}^{18}$F, ${}^{99m}$Tc
Black et al., 2021 [71]	Lungs	Unknown	Not yet	N/A

**Table 2 diagnostics-11-02333-t002:** Examples of information which should be included in future publications of 3D-printed phantoms.

Imaging	Modality	
	Processing	
Software	Application in workflow	
3D Printer	Model	
	Material	Type
		Relevant properties
Technical	Layer thickness	
	Phantom thickness	
	Attachments	Type
		Position
	Filling method	
	Assembly	Single/multiple parts
		Assembly method
Key design choices		
Flow chart of the design process

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
