# Peer review of "Absolute Quantification in Diagnostic SPECT/CT: The Phantom Premise"

_diagnostics, 2021, doi:10.3390/diagnostics11122333_

Round 1

Reviewer 1 Report

This is a well structured review and would help those with an interest in using 3D printed phantoms for quantification in SPECT and PET.  It only requires very minor modifications

One missing reference is to the EARL website

One suggestion to maybe consider is to appeal for a central database of phantoms already manufactured so that other groups can make their own.  This would obviously need permissions from the developing sites but would be a useful asset if under the auspices of a central body such as EANM/EFOMP etc.

Author Response

We would like to thank the reviewer for the thoughtful comments and suggestions.

Reference to the EARL website was added

We have added a suggestion for the establishment of an online repository such as NIH 3D Print Exchange.

Reviewer 2 Report

This is a well written review article on 3D phantoms that helpfully summaries the current status of the field which I have minimal comments on.  Potentially in the discussion the authors could explicitly add what the authors would suggest would be necessary method information to be included in future 3D phantom papers.

Minor comments:

Line 119 add Monte Carlo collimator modelling as researchers have used this and it is available commercially

Line 149 has a ? in ref

Line 252 ref 58 missing hyperlink

Line 322 typo “shown”

Author Response

We would like to thank the reviewer for the thoughtful comments and suggestions.

We have added a small part in the discussion with a table for suggested information to be included in publications on 3D printed phantoms.

Added the reference related to MC collimator modelling.

Fixed the reference

Added hyperlink

Fixed the typo